# TEXTBIND: MULTI-TURN INTERLEAVED MULTIMODAL INSTRUCTION-FOLLOWING IN THE WILD

## ABSTRACT

Large language models with instruction-following abilities have revolutionized the field of artificial intelligence. These models show exceptional generalizability to tackle various real-world tasks through their natural language interfaces. However, their performance heavily relies on high-quality exemplar data, which is often difficult to obtain. This challenge is further exacerbated when it comes to multimodal instruction following. We introduce TEXTBIND, an almost annotation-free framework for empowering LLMs with multi-turn interleaved multimodal instruction-following capabilities. Our approach requires only image-caption pairs and generates multi-turn multimodal instruction-response conversations from a language model. To accommodate interleaved image-text inputs and outputs, we devise MIM, a language model-centric architecture that seamlessly integrates image encoder and decoder models. Extensive quantitative and qualitative experiments demonstrate that MIM trained on TEXTBIND achieves remarkable generation capability in multi-modal conversations compared to recent baselines.

## 1 INTRODUCTION

Artificial intelligence (AI) has experienced a significant paradigm shift with the rise of large language models (LLMs). These models are capable of processing a wide range of natural language processing (NLP) applications through natural language interactions with users (OpenAI, 2022; 2023). Despite their remarkable performance, these models cannot process and generate visual content.

Recently, a number of efforts have been made to augment LLMs with visual perception and understanding abilities. Prior work uses template-based instruction-following datasets for training (Xu et al., 2023b; Dai et al., 2023; Li et al., 2023c). These datasets comprise a variety of classic computer vision (CV) tasks, e.g., object detection, with each task being converted into an instructional format using a handful of human-written natural language instructions. However, classic CV tasks often represent manageable and focused abstractions or simplifications of real-world tasks (Marr, 2010), they generally fall short in representing the true variety and complexity of real-world tasks and capturing the lexical diversity of human language. For example, most of them are single-turn inquiries about a single input image, whereas a small fraction supports multi-turn textual interactions or multiple image inputs. Consequently, the instruction-following capabilities of models trained on these datasets remain limited in open-world scenarios (Xu et al., 2023a). This is reminiscent of the early development of instruction tuning in NLP, where public NLP tasks were eventually superseded by high-quality, diverse open-world instruction data (Ouyang et al., 2022). Nevertheless, collecting such data for multimodal models can be extremely costly.

In this paper, we address the above challenge by introducing TEXTBIND, an almost annotation-free framework for augmenting LLMs with multi-turn interleaved multimodal instruction-following capabilities. The main idea is to represent images through their textual descriptions, e.g., captions, and utilize an LLM to generate multi-turn instructions and responses. To ensure the coherence and meaningfulness of the constructed multi-turn conversations, we propose a series of strategies such as topic-aware image sampling and human-in-the-loop refinement of in-context demonstrations. TEXTBIND can harvest large-scale datasets given the abundance of public image-caption pairs.

TEXTBIND provides examples of processing and generating arbitrarily interleaved image-and-text content. To accommodate interleaved image-text inputs and outputs, we devise MIM, a multimodal model that emphasizes the reasoning abilities of LLMs and seamlessly integrates image encoder and

decoder models. The comparison of TEXTBIND and previous representative datasets is shown in Tab. 8 (Appx. D), accompanied by an illustration of the models trained on different datasets in Fig. 10 (Appx. D).

To assess the generative capabilities of MIM trained on TEXTBIND, we perform comprehensive analyses in the context of multi-modal conversations (§6). In particular, thorough reference-based automatic evaluation metrics reveal that the MIM model substantially surpasses MiniGPT-4 Zhu et al. (2023) and LLaVA Liu et al. (2023a) in textual response generation, and outperforms GILL Koh et al. (2023a) and Stable Diffusion Podell et al. (2023) in image generation by a considerable margin. Furthermore, our holistic evaluation demonstrates that MIM consistently outperforms the representative baselines. In addition, our qualitative experiments show that MIM trained on TEXTBIND can perform a wide range of tasks, including composing engaging stories inspired by a set of images (Fig. 10), comparing the common and different parts in multiple images (Fig. 6b (Appx. A)), explaining concepts with vivid images (Fig. 5a (Appx. A)), generating long coherent stories with illustrations (Fig. 4 (Appx. A)), etc. More demonstrations are shown in Appx. A. Most interestingly, the core innovation of our model is its capability to interact with users naturally. For instance, rather than requiring users to supply the model with explicit descriptions of the desired image, our model can spontaneously generate images in proper conversation contexts. We hope TEXTBIND serves as an initial step towards building AGI that can interact with humans flexibly in different modalities and broad real-world scenarios.

## 2  RELATED WORK

**Multimodal Datasets**    Existing multimodal datasets can be broadly classified into two categories: (1) Conventional datasets for specific vision-language tasks such as image captioning (Chen et al., 2015; Agrawal et al., 2019; Young et al., 2014) and visually-grounded question answering (Hudson & Manning, 2019; Marino et al., 2019; Singh et al., 2019; Lu et al., 2022; Zhou et al., 2018; Goyal et al., 2017; Gurari et al., 2018). (2) Recent dataset for general instruction following. For instance, MultiInstruct (Xu et al., 2023b), InstructBLIP (Dai et al., 2023), and M3IT (Li et al., 2023c) convert existing vision-language datasets into a unified instructional format with handcrafted templates. This approach is reminiscent of the early explorations on instruction tuning in NLP (Wei et al., 2022; Sanh et al., 2022), where existing NLP tasks were phrased as instructions. However, it has been reported that such instruction-tuned multimodal models still generalize poorly to open-world scenarios (Xu et al., 2023a). This finding also aligns with the observations in NLP (Ouyang et al., 2022), where template-based instruction tuning is less effective than instruction tuning data collected from real-world scenarios due to its restricted diversity. There are also some attempts to convert the output of existing vision-language models into natural language answers for constructing instruction-tuning data (Liu et al., 2023a; Zhu et al., 2023; Chen et al., 2023a).

Compared to existing instruction-tuning data, the examples in TEXTBIND (1) generally exhibit greater task and lexicon diversity; (2) typically involve multiple images scattered throughout a multi-urn conversation; (3) support multimodal output (image generation).

**Multimodal Models**    To augment existing LLMs with visual abilities, one straightforward approach is to employ off-the-shelf vision models as external tools. That is, the LLM calls expert vision models through their language interfaces for completing specific visual tasks when needed (Wu et al., 2023; Shen et al., 2023; Chen et al., 2023b; Zou et al., 2022; Yang et al., 2023; Surís et al., 2023).However, these approaches may suffer from cross-modal information loss and lack of generality.

Recently, end-to-end multimodal language models have garnered significant interest. Flamingo (Alayrac et al., 2022) and OpenFlamingo (Alayrac et al., 2022) are among the pioneering work to extend LLMs to vision-language pretraining. Different from training from scratch, subsequent research efforts have focused on integrating pretrained vision and language models. BLIP-2 (Li et al., 2023b) proposes Qformer to align the feature spaces of vision models and language models. To date, various network architectures and training strategies have been proposed (Zhu et al., 2023; Liu et al., 2023a; Ye et al., 2023; Li et al., 2023a; Zhang et al., 2023; Du et al., 2022; Chen et al., 2023a; Dai et al., 2023). However, these models are limited to the use of visual content as input. Our work is inspired by recent work on LLM-empowered image retrieval or generation (Koh et al., 2023b;a) and the pioneer work of (Sun et al., 2022) for chitchat in the context of single photo sharing. Contrary

to prior work, we aim to present the first instruction-following model capable of processing and generating arbitrarily interleaved image-text inputs and outputs.

**Evaluation**   Conventional vision datasets designed for specific tasks and scenarios may suffer from data contamination issues for evaluating LLMs. Recently, efforts have been made to provide systematic evaluations with a broader coverage of diverse visual abilities. MME (Fu et al., 2023) is an evaluation dataset containing visually-grounded Yes/No questions. OwlEval (Ye et al., 2023) is a benchmark comprising 82 questions based on 50 images and relies on human feedback evaluation. The test size is limited, and the results may suffer from subjective bias. In response to these challenges, MMbench (Liu et al., 2023b) and MM-Vet (Yu et al., 2023) are two recent benchmarks aiming to offer more comprehensive evaluations by incorporating the use of ChatGPT/GPT4 for answer verification. LVLM Arena (Xu et al., 2023a), an online evaluation framework that ranks different models using human judgment, is also introduced. However, the above benchmarks primarily focus on question answering based on a single image at the beginning of a conversation.

## 3   TEXTBIND

In this work, we seek to enhance the multi-turn instruction-following capabilities of a language model in the context of arbitrarily interleaved images and text. Constructing such datasets poses significant challenges: 1) it demands inventive thinking for devising high-quality visually-grounded instructions and their responses; 2) it requires specialized expertise to craft appropriate images. To tackle these issues, we introduce TEXTBIND, a method that predominantly resorts to existing *text-only language models*[1] to produce the desired data.

### 3.1   DEFINITION OF DATA

The goal of TEXTBIND is to construct a collection of multi-turn conversation such as $[\boldsymbol{x}_u^1, \boldsymbol{x}_a^1, \ldots, \boldsymbol{x}_u^T, \boldsymbol{x}_a^T]$, where $T$ is the number of turns, $\boldsymbol{x}_u^i$ denotes the $i$-th instruction from the user, and $\boldsymbol{x}_a^i$ represents the $i$-th response from the assistant. The conversation is also accompanied by an image set $\{\boldsymbol{m}_1, \ldots, \boldsymbol{m}_n\}$, where $n$ is the number of unique images in this conversation. Each instruction $\boldsymbol{x}_u^i$ or response $\boldsymbol{x}_a^i$ is a sequence of tokens in $\mathcal{V}_{\text{lang}} \cup \mathcal{V}_{\text{img}}$, where $\mathcal{V}_{\text{lang}}$ is the ordinary vocabulary of a language model and $\mathcal{V}_{\text{img}}$ contains $n$ distinct pointers to the images $\boldsymbol{m}_1, \ldots, \boldsymbol{m}_n$ respectively. It is worth noting that every image can appear at any point within the conversation.

### 3.2   AUTOMATIC DATA GENERATION

TEXTBIND consists of a three-step pipeline: 1) topic-aware image sampling for ensuring the coherence of each conversation and the diversity across conversations; 2) LLM-empowered multi-turn instruction-response generation to create natural and practical conversations; 3) post-processing and filtering to eliminate low-quality data. An overview of the TEXTBIND pipeline is shown in Fig. 1.

**Topic-Aware Image Sampling**   The initial step of TEXTBIND entails assembling groups of images that will serve as the foundation for generating multi-turn conversations. In order to facilitate coherent, meaningful, and practical conversations, the images within each group should exhibit meaningful interconnections. Furthermore, to guarantee a comprehensive representation of real-world scenarios, the topics of images across different conversations should demonstrate a wide range of diversity.

Following the above inspirations, we employ unsupervised clustering algorithms to group the images in our dataset into clusters and execute a two-step image sampling process for each conversation. Concretely, we use the image encoder of the CLIP model (Radford et al., 2021) to obtain vector representations of images. Then, we execute the $k$-means algorithm to classify all images into $K$ clusters (topics). Examples of such clusters are given in Fig. 1. For each conversation, we randomly sample a cluster from the available $K$ clusters, then sample $n \in \{2, 3, 4\}$ images from the chosen cluster. We want to higlight that the clustered images are semantically relevant, rather than visually similar.

---

[1]Although OpenAI claims that GPT4 supports visual input, this feature is yet to be made public.

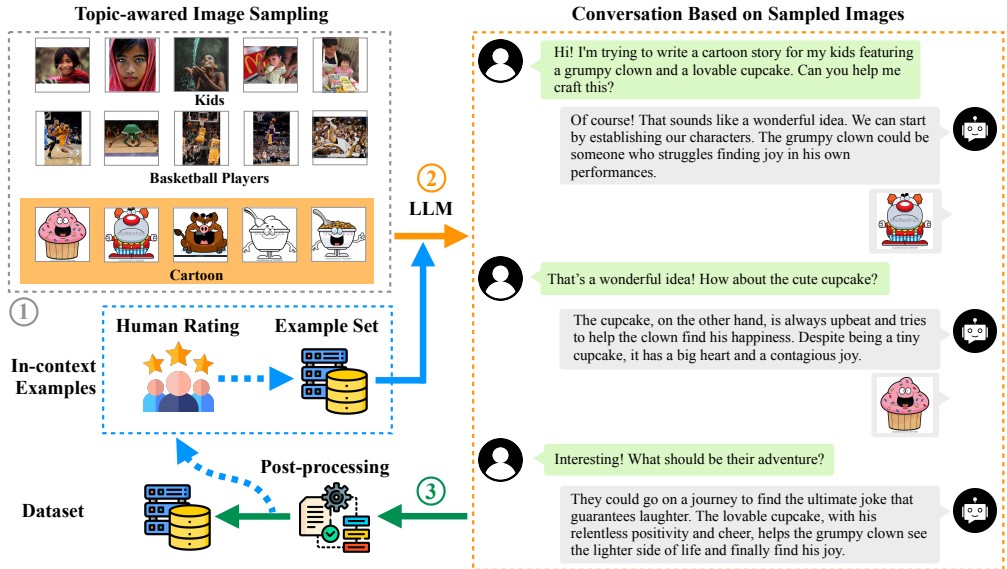

Figure 1: Illustration of the TEXTBIND method. In the top-left corner, we display five representative images from each of the three example clusters obtained via unsupervised clustering. On the right-hand side, a conversation is showcased and constructed using two randomly sampled images from the cartoon cluster. In the bottom-left corner, we outline the additional TEXTBIND pipeline, which includes human-in-the-loop refinement and post-processing stages.

**Generation of Multi-turn Conversations**    After selecting a list of images, we proceed to leverage a text-only LLM, such as GPT-4, to simulate a conversation between a user and an assistant based on the chosen images. The core idea is to let LLMs receive and process the textual descriptions of the images as if they see the actual images. Given the abundance of publicly available image-caption pairs, we propose representing an image with an XML-like string `<imgX> DESCRIPTION </imgX>`, where `DESCRIPTION` serves as a placeholder for the image caption, `<imgX>` and `</imgX>` mark the caption boundaries, and `X` denotes the image index in the input image list. After generating the conversation, we replace the XML-like strings in the conversation with the original images. Importantly, to ensure that a caption faithfully describes its corresponding image, we employ the CLIP model (Radford et al., 2021) to filter out image-caption pairs with matching scores below a high threshold.

The detailed prompt can be found in Appx. B, and examples of generated conversations before mapping the textual descriptions back to visual images are shown in Appx. C. In the prompt, we also provide in-context examples to improve the generation quality. We collect the in-context examples through a human-in-the-loop refinement process, which is elaborated in §3.3.

**Post-processing and Low-quality Filtering**    To ensure data quality, we filter out conversations where there is a pair of input and output image descriptions with an edit distance higher than 0.1. We also exclude conversations containing image descriptions not present in the provided image list and conversations containing formatting errors such as co-reference errors and invalid image tags.

### 3.3   HUMAN-IN-THE-LOOP REFINEMENT

In-context learning has been demonstrated to be crucial for enhancing the generation quality of LLMs (Brown et al., 2020; Wang et al., 2023). Therefore, we also construct a seed set of high-quality in-context examples $\mathcal{S}$. The seed set $\mathcal{S}$ begins as an empty set and is iteratively updated with human feedback. In each iteration, we follow the steps detailed below:

1. We employ the latest $\mathcal{S}$ and the template in Appx. B, and generate 100 new conversations using TEXTBIND (§3).

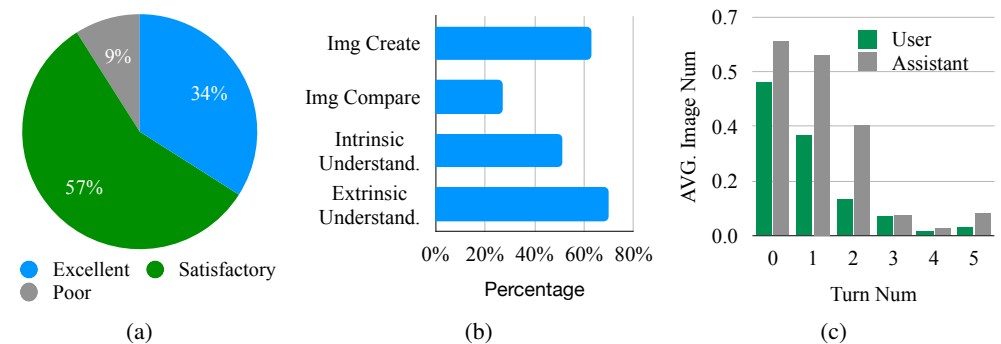

Figure 2: Statistics of data quality and diversity. The results in Fig. 2a and 2b are based on the human annotations on 100 randomly sampled conversations.

2. We manually analyze the generated conversations. Each conversation is assigned a quality label ("Excellent", "Satisfactory", or "Poor"). Besides, we label the visual abilities required for each conversation. The detailed annotation guideline for quality labels and visual abilities is outlined in Tab. 9 (Appx. E).

3. We add the generated conversations with "Excellent" or "Satisfactory" labels to $\mathcal{S}$.

To ensure diversity among different conversations, we randomly sample three in-context examples from the seed set for each generation. We further require that at least one in-context example is labeled "Excellent" and the three sampled examples encompass all four visual abilities. After three iterations, we fix the seed set and employ it to generate the remaining data. The percentage of "Poor" data annotated by humans declines from 30% to 9%.

| Statistics | |
|---|---|
| # of conversations | $25,629$ |
| Avg. # turns in conversations | $3.36$ |
| Avg. # images | |
| in conversations | $2.46$ |
| in instructions | $0.94$ |
| in responses | $1.52$ |
| Avg. # words | |
| in conversations | $285.90$ |
| in instructions | $78.66$ |
| in responses | $207.24$ |

Table 1: Statistics of the dataset by applying TEXTBIND to GPT-4.

## 4 TEXTBIND DATA FROM GPT4

We apply TEXTBIND to GPT4 and the CC3M dataset (Sharma et al., 2018; Changpinyo et al., 2021) as a case study. The details of the construction process can be found in Appx. F. In this section, we present comprehensive analyses of the constructed dataset. The constructed dataset will be released.

**Statistics** As depicted in Tab. 1, our constructed dataset comprises $25,629$ conversations. The average number of turns per conversation is $3.36$ (each turn is defined as a pair of instruction and response). The mean number of images in each conversation is $2.46$.

**Diversity** To understand the lexical and task diversity of our constructed data, we identify four types of required visual abilities and show their distribution in Fig. 2b. We observe that a significant portion of conversations in our dataset focuses on more insightful and informative tasks, such as extrinsic understanding and image comparison. For topic diversity, we display three randomly sampled clusters in Fig. 1. The distribution of images across different turns is depicted in Fig. 2c. We also compare the lexical diversity of our dataset and existing datasets in Tab. 2.

| Dataset | Instruct | Response | Overall |
|---|---|---|---|
| LLAVA | 1.56 | 1.84 | 1.70 |
| MINIGPT-4 | 0.00 | 1.11 | 0.89 |
| MULTIINSTRUCT | 0.51 | 1.69 | 0.51 |
| PLATYPUS | 0.98 | 0.75 | 0.78 |
| SHIKRA | 0.89 | 1.08 | 0.87 |
| TEXTBIND | **1.76** | **1.92** | **1.84** |

Table 2: Averaged diversity scores of roles in various datasets. Details of this analysis are in Appx. D.

**Quality** To check the quality of the generated data, we randomly sample 100 conversations and perform an in-depth error analysis. As shown in Fig. 2a, only $9\%$ conversations in the dataset are labeled as "Poor". Note that we label the whole conversation as "Poor" if any of its turns has a

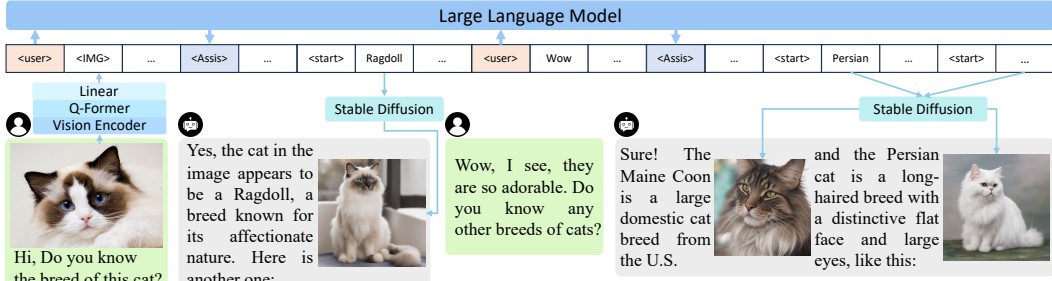

Figure 3: The architecture of MIM. It integrates a vision model, a language model, and a stable diffusion model. MIM is able to process multi-turn interleaved multimodal inputs and outputs.

problem. We analyze the error types (image-caption mismatch, incoherence, and hallucination) in Appx. G.

# 5 AUGMENTING LLMs WITH VISUAL I/O

## 5.1 MODEL

To support interleaved multimodal inputs and outputs, we supplement LLMs with visual input and output modules. Specifically, LLama2-Chat[2] (Touvron et al., 2023) is employed as the backbone LM. For visual input, we use the vision encoder from BLIP2 (Li et al., 2023b)[3], followed by a pretrained Q-Former model (Li et al., 2023b) that maps the features from the vision model into the embedding space of the LM. Inspired by GILL (Koh et al., 2023a), we attempt to learn a mapping from the output space of the LM to the input space of a stable diffusion (SD) model (Rombach et al., 2022) (in this work, the embeddings produced by the text encoder of Stable Diffusion XL (Podell et al., 2023)). To this end, we explore three model variants in our preliminary experiments. The training examples of the MIM model follow the standard of Llama-Chat, as shown in Appx. J. The content in different conversation turns is concatenated. The model is trained to minimize the cross-entropy loss on the assistant's turns, conditioned on the entire preceding conversation history.

**Q-Former as Medium**. We add a special token  to the vocabulary of the LM, indicating that an image should be generated when it is emitted. We then use a Q-Former (Li et al., 2023b) that takes all previous hidden states of the LM as input and outputs the SD embeddings.

**Q-Former with Prompt Tokens as Medium**. To further leverage the reasoning abilities of the LM, we incorporate a series of special tokens (<img1>, ..., ), instead of a single token (), to the LM. When <img1> is emitted, the generation of the special token sequence is enforced, serving as additional reasoning steps for predicting the forthcoming image. Subsequently, the Q-Former only accepts the hidden states of special tokens as input.

**Language Description as Medium**. The previous two variants try to align the continuous hidden spaces of different models. An alternative is to use discrete language descriptions for information exchange, as depicted in Fig. 3. Specifically, we add two special tokens, <start> and <end>, and encode the generated text between these two tokens using the text encoder in the SD model.

Similar to GILL (Koh et al., 2023a), we optimize the first two variants by minimizing the mean squared error (MSE) loss between the output embeddings and the SD embeddings. For the third variant, we employ the standard cross-entropy loss. We empirically find that only the last method demonstrates satisfactory performance on **m**ulti-turn **i**nterleaved **m**ultimodal instruction-following, for which we name it MIM.

---

[2]https://huggingface.co/meta-llama/Llama-2-7b-chat-hf
[3]https://huggingface.co/Salesforce/blip2-flan-t5-xxl

## 5.2 TRAINING

Our training process consists of two stages, namely, the multimodal alignment stage and the multimodal instruction tuning stage.

**Multimodal Alignment** The first stage aims to align the feature spaces of the vision model and the language model. We utilize massive image-caption pairs for training, drawing from datasets such as Conceptual Captions (Changpinyo et al., 2021; Sharma et al., 2018) and SBU (Ordonez et al., 2011). During training, only the Q-Former connecting the vision and language models is optimized while other model components remain frozen.

**Multimodal Instruction Following** The second stage further trains the joint model on multimodal instruction tuning data to improve its instruction-following capabilities. The Q-Former model and LLM are optimized in this stage. In addition to TEXTBIND data, we also explore existing multimodal instruction data including MultiInstruct (Xu et al., 2023b), MiniGPT-4 (Zhu et al., 2023), LLaVA (Liu et al., 2023a), and Shikra (Chen et al., 2023a).

## 6 EXPERIMENTS

To verify the effectiveness of the proposed methods, we carry out quantitative evaluations against a set of recent baselines. Our quantitative evaluations are divided into three parts: textual response generation, image generation, and a holistic evaluation of multimodal instruction-following.

### 6.1 TEXTBINDEVAL

To facilitate comprehensive and dedicated evaluation for instruction-following in realistic scenarios, we construct a new dataset named TEXTBINDEVAL. TEXTBINDEVAL is initially generated through the automatic pipeline of TEXTBIND (§3) and subsequently refined by human annotators. These annotators are tasked with discarding low-quality examples or rectifying amendable issues such as revising incoherent or hallucinated content. After a rigorous review, we establish an evaluation dataset comprising 278 conversations in total.

### 6.2 TEXTUAL RESPONSE GENERATION

**Setup** We consider each assistant turn of each conversation in TEXTBINDEVAL as a test point. All its preceding context is treated as input (which may contain interleaved images and text), and the goal is to generate a coherent and helpful response. We measure the response quality using a set of reference-based evaluation metrics such as BLEU (Papineni et al., 2002), ROUGE (Lin, 2004), and BERTScore (Zhang et al., 2020). We also report the Diversity (Su et al., 2022) scores of the generated responses. For simplicity, we replace any image in the responses with a special token `<image>`.

For a fair comparison, we compare different MIM models trained on different datasets (Xu et al., 2023b; Zhu et al., 2023; Liu et al., 2023a; Chen et al., 2023a)[4] and GILL (Koh et al., 2023a)[5]. The implementation details are shown in Appx. H.

**Results** As shown in Tab. 3, the MIM model trained on TEXTBIND outperforms all other baselines by wide margins across all evaluation metrics. The results suggest that more realistic and diverse training data such as TEXTBIND is necessary for tackling open-world tasks, which cannot be well-supported by existing template-based and VQA-like datasets. Nevertheless, we also find that the performance can be further improved when combining different datasets, indicating that there is a complementary relationship between TEXTBIND and existing datasets.

### 6.3 IMAGE GENERATION

**Setup** The models trained on existing datasets, i.e., the baselines in §6.2 except for GILL, are incapable of generating images. To showcase the image generation capabilities of our model, we compare it with Stable Diffusion XL (SD-XL) (Podell et al., 2023) and GILL (Koh et al., 2023a). In

---

[4]The original papers of these datasets used distinct model architectures such as different pretrained language models. One common feature is that all of them do not support image generation.

[5]For a fair comparison, we replicate GILL using the same image-captioning data to train by our models.

| Methods | BLEU-2 | BLEU-4 | ROUGE-2 | ROUGE-L | BERTScore | Diversity |
|---|---|---|---|---|---|---|
| GILL (Koh et al., 2023a) | 3.97 | 1.44 | 4.61 | 13.97 | 0.847 | 0.902 |
| MultiInstruct (Xu et al., 2023b)[6] | 7.16 | 2.27 | 3.16 | 10.60 | 0.830 | 0.654 |
| MiniGPT-4 (Zhu et al., 2023) | 9.24 | 3.29 | 6.77 | 17.56 | 0.858 | 0.658 |
| LLaVA (Liu et al., 2023a) | 12.16 | 4.41 | 8.66 | 19.79 | 0.872 | 0.852 |
| Shikra (Chen et al., 2023a) | 10.37 | 3.83 | 7.79 | 18.63 | 0.864 | 0.722 |
| TEXTBIND | **24.45** | **11.83** | **15.45** | **28.69** | **0.891** | **0.927** |
| Mix | 27.64 | 14.49 | 17.90 | 31.22 | 0.896 | 0.912 |

Table 3: Evaluation of textual response generation. Mix represents the mixture of MultiInstruct, MiniGPT-4, LLaVA, Shikra, and TEXTBIND.

| Model | Instruction-following | Multimodal Context Understanding | Informativeness |
|---|---|---|---|
| LLaVA (Liu et al., 2023a) | 3.59 | 3.56 | 3.78 |
| TEXTBIND | 3.99 | 3.82 | 3.72 |

Table 5: Fine-grained analysis using human evaluation.

addition, we present the results of the two model variants described in §5.1, namely, **Q-former as Medium** and **Q-former with Prompt Tokens as Medium**.

We take each image from the assistant in TEXTBINDEVAL as a test point. All its preceding context is taken as input, and the models are enforced to output an image. We take the original images in TEXTBINDEVAL as references. Following Koh et al. (2023a), we evaluate image generation with two reference-based metrics: (1) **CLIP Similarity**. We use the CLIP vision encoder to produce image representations and compute the cosine similarity between generated images and reference images. A higher score means better semantic similarity. (2) **Learned Perceptual Image Path Similarity (LPIPS)**. LPIPS (Zhang et al., 2018) measures the distance between generated images and reference images. A lower score means that images are more similar in perceptual space. (3) **Frechet Inception Distance** (FID). FID measures the distributional difference between the generated images and reference images. A lower score indicates better resemblance to reference images.

**Results** To gain further insights into the multi-turn instruction-following abilities, we group different test points by the number of previous conversation turns. The results are shown in Tab. 6. As seen, MIM generally achieves better performance than SD-XL and GILL across different turns and evaluation metrics. Importantly, the performance gaps are enlarged as the number of turns increases. This indicates that our model exhibits a better understanding ability of multi-turn conversations. Compared to the two model variants, MIM is substantially better. Our case study reveals that the disparity stems from the *one-to-many* nature of image generation in real-world conversations. Unlike generating images for explicit descriptions, there can exist numerous distinct images for a given conversation context. Operating in the hidden space may inadvertently average all possibilities, resulting in ambiguous or noisy images. However, MIM mitigates the *one-to-many* issue by taking full advantage of the autoregressive generation of language models for decision-making.

## 6.4 HOLISTIC EVALUATION

In addition to the above automatic evaluation, we also conduct a holistic evaluation of instruction-following abilities through human annotation. To further show where the derived dataset and training helps, we ask human annotators to evaluate the quality of the generated responses in terms of three fine-grained dimensions: instruction-following (fulfill the intent of users), multi-modal context understanding (correctly understand the information in text and images), and the informativeness of the generated responses. For each dimension, a human annotator will assign a score in $\{1, 2, 3, 4\}$. The four scores ranging from 1 to 4 indicate "major error", "minor error", "acceptable", and "perfect", respectively. We compare TEXTBIND with LLaVA (the second best model in our holistic evaluation in 6) on 100 randomly sampled data. As shown in the Table 5, the model trained on TEXTBIND can better follow the instructions of humans and leverage the multi-modal context. Notably, the informativeness of model trained on TEXTBIND is comparable with that trained on LLaVA.

| Model | CLIP Similarity (↑) | | | LPIPS (↓) | | | FID (↓) |
|---|---|---|---|---|---|---|---|
| | Turn-1 | Turn-2 | Turn-3 | Turn-1 | Turn-2 | Turn-3 | All |
| SD-XL (Podell et al., 2023) | 0.612 | 0.599 | 0.608 | **0.712** | 0.735 | 0.735 | 144.76 |
| GILL (Koh et al., 2023a) | 0.569 | 0.550 | 0.530 | **0.712** | 0.734 | 0.742 | 158.64 |
| Q-Former as Medium | 0.558 | 0.568 | 0.592 | 0.717 | 0.728 | 0.729 | 155.01 |
| Q-Former with Prompt Tokens as Medium | 0.566 | 0.571 | 0.606 | 0.718 | 0.727 | 0.732 | 152.23 |
| MIM | **0.640** | **0.645** | **0.673** | **0.712** | **0.720** | **0.726** | **139.46** |

Table 6: Evaluation of image generation.

| Training Dataset | MME | | MMBench | | | | | | | MM-Vet |
|---|---|---|---|---|---|---|---|---|---|---|
| | Perception | Cognition | LR | AR | RR | FP-S | FP-C | CP | Overall | - |
| MultiInstruct (2023b) | **1099.16** | **302.50** | 11.93 | 39.79 | 28.64 | 28.75 | 23.20 | 41.91 | 31.54 | 17.2 |
| MiniGPT-4 (2023) | 0.00 | 0.00 | **14.20** | 50.52 | 17.37 | 32.75 | 15.20 | 41.70 | 31.87 | 9.8 |
| LLaVA (2023a) | 683.28 | 267.86 | 7.95 | 55.71 | 31.46 | 42.50 | 31.60 | **56.60** | 42.10 | 23.4 |
| Shikra (2023a) | 166.87 | 2.86 | 18.18 | **64.01** | 22.54 | 39.75 | 31.20 | 50.43 | 41.10 | 19.9 |
| TEXTBIND | 549.00 | 226.43 | 11.93 | 36.33 | 6.57 | 23.25 | 6.00 | 33.83 | 22.64 | 19.4 |
| Mix | 1023.33 | 255.00 | 13.64 | 56.75 | **37.09** | **43.50** | **42.80** | 55.32 | **44.94** | **23.9** |

Table 7: Results on MME (Fu et al., 2023), MMBench (Liu et al., 2023b), MM-Vet (Yu et al., 2023).

**Setup** We randomly sample 100 contexts from TEXTBINDEVAL and evaluate the responses generated by MIM and two representative baselines, LLaVA (Liu et al., 2023a) and GILL (Koh et al., 2023a). We instruct three human annotators to score the quality of each generated response on a Likert scale from 1 to 4 (The details of evaluation guideline are in Appx. I).

| Methods | AVG. Score | Percent. (≥ 3) |
|---|---|---|
| GILL | 1.71 | 0.19 |
| LLaVA | 2.93 | 0.89 |
| MIM | 3.39 | 0.70 |

Table 4: Averaged human scores and the percentage of averaged scores ≥ 3. Krippendorff's $\alpha = 0.75$.

**Results** As shown in Table 4, MIM achieves higher human scores than GILL and LLaVA, indicating its remarkable generation capability in open-world multi-modal conversations. In addition, the Krippendorff's $\alpha = 0.75$ indicates a high inter-annotation agreement between annotators.

## 6.5 RESULTS ON EXISTING BENCHMARK

Finally, we report the results on two popular multimodal benchmarks, MME (Fu et al., 2023), MMBench (Liu et al., 2023b), and MM-Vet (Yu et al., 2023). As shown in Tab. 7, TEXTBIND gets a relatively lower score than other datasets. The reason stems from the intrinsic difference between TEXTBIND and the two benchmarks. TEXTBIND focuses more on realistic instructions (e.g., create a story based on the images, give some suggestions for having fun in the winter). In contrast, MME, MMBench and MM-Vet focus more on VQA questions, e.g., who is this person, what is the color of the object, which are more similar to the data in MultiInstruct, LLaVA, and Shikra. For example, the model trained on MultiInstruct achieves the best performance on MME, though it displays the worst performance in open-world scenarios in Tab. 3. Another interesting observation is that the mix of all datasets attains the best overall performance on MMBench, indicating that different datasets are complementary. In other words,the capabilities that TextBind can bring are almost orthogonal to existing multimodal instruction-following datasets.

## 7 CONCLUSION

In conclusion, the introduction of the TEXTBIND framework has opened new doors for enhancing large language models with multi-turn interleaved multimodal instruction-following capabilities. By requiring only image-caption pairs, our approach significantly reduces the need for high-quality exemplar data, making it a more accessible and scalable solution for various real-world tasks. The MIM architecture seamlessly integrates image encoder and decoder models, enabling the model to effectively handle interleaved image-text inputs and outputs. Comprehensive quantitative and qualitative experiments demonstrate the remarkable performance of MIM, trained on TEXTBIND, when compared to recent baselines in open-world multimodal conversations.

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

## A    DEMONSTRATIONS

The four high-level characteristics of the TEXTBIND dataset equips MIM with a variety of capabilities. We demonstrate those capabilities with concrete user cases.

**Image Creation**    One core innovation of TEXTBIND is that it enables the model to create images based on the conversation context without explicit dictations from the users. This characteristic is extremely useful for open-world scenarios, because in many cases people may just have an implicit intention and have no clear thoughts about what the images should be. We observe that our model can explain concepts and ideas for users with vivid images (Figure 5a), creating images with correct emotions (Figure 5b), and editing images based on the whole context (Figure 5c and 5d). Furthermore, as shown in Figure 4, we discover that our model is proficient in generating long stories featuring interleaved text and images while maintaining exceptional coherence.

**Image Comparison**    Another interesting feature of TEXTBIND is that it can compare or relate the information in multiple images. For example, our model can correctly explain the different and common parts between images in Figure 6.

**Intrinsic & Extrinsic Image Understanding**    The model trained on TEXTBIND can understand the content in images precisely in a multi-turn conversation. In all the three sub-figures of Figure 7, the model precisely follows the human instructions and explains the details of the images to users. Moreover, TEXTBIND also enables the model to explore the meaning of an image beyond the symbols in it. For example, the model also explains the the influence of Bob Dylan's album in Figure 7b and the impact of iPhone in Figure 7c.

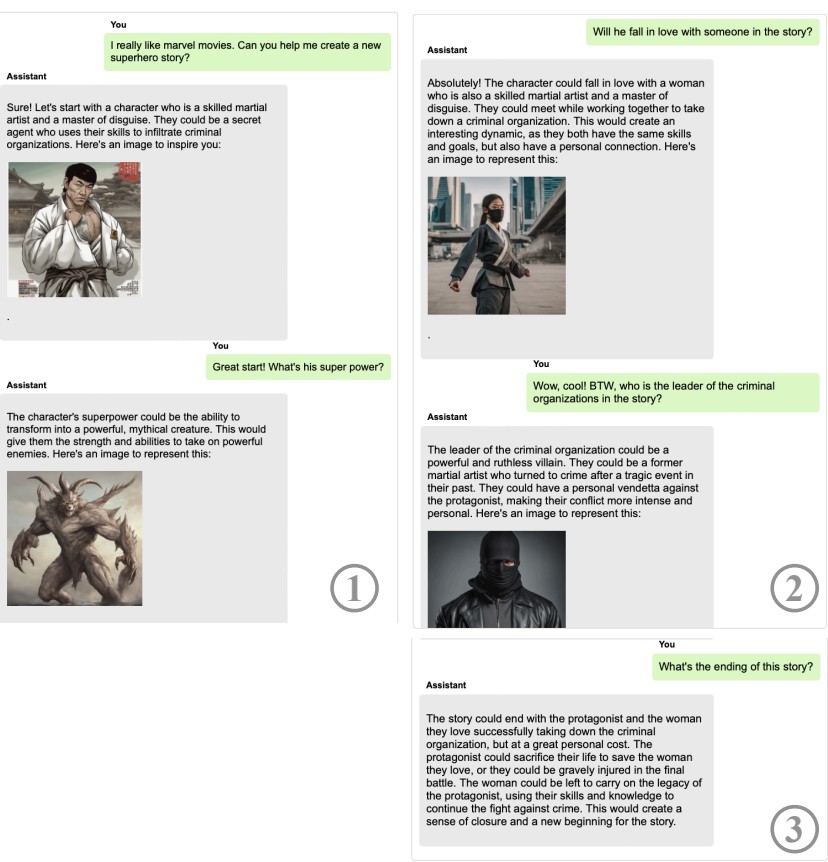

Figure 4: Generation of a long story with interleaved text and images.

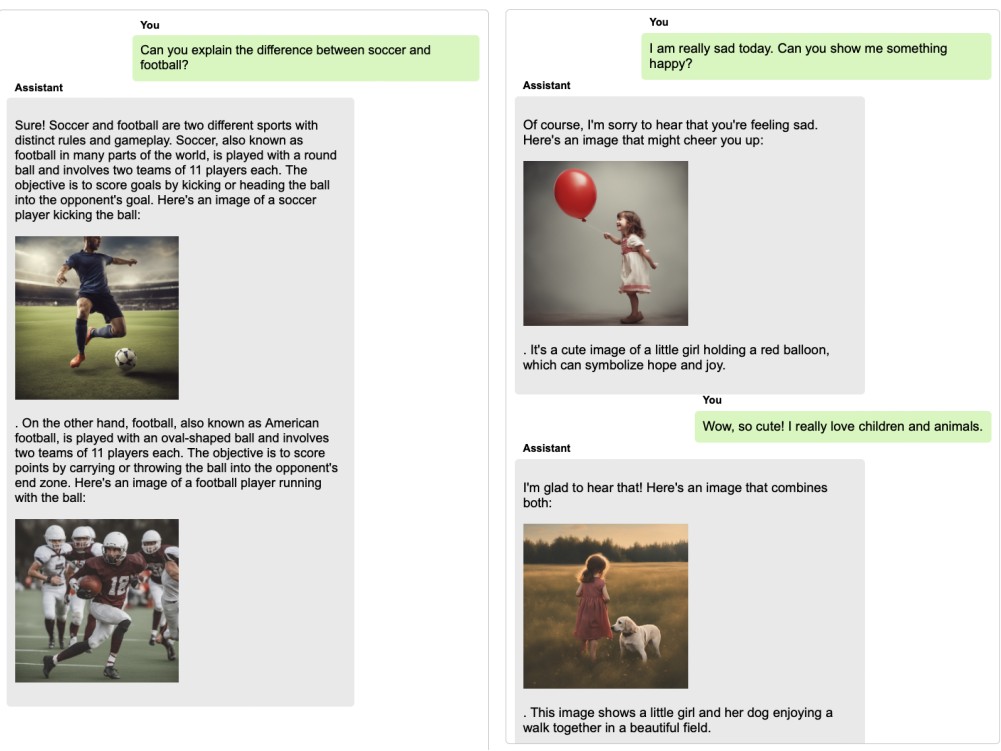

(a) Explaining concepts with multiple images.     (b) Creating images with correct emotions.

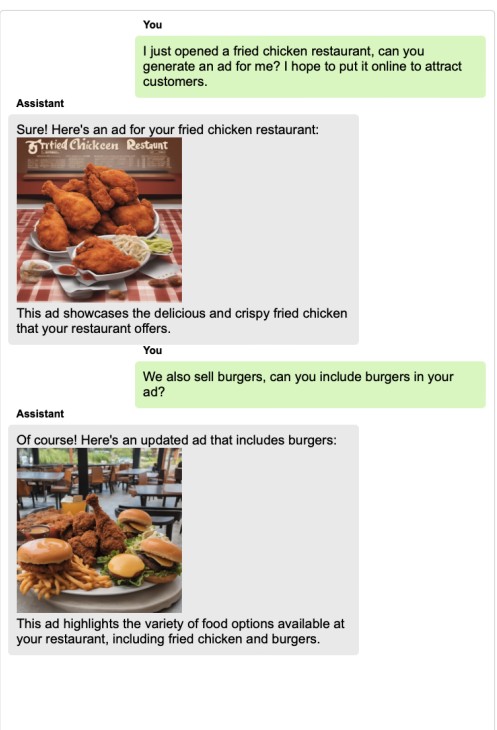
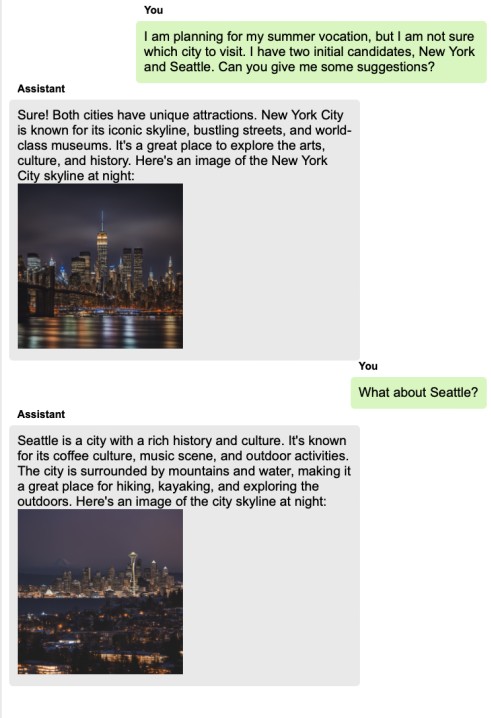

(c) Editing images based on context.     (d) Creating images based on context.

Figure 5: User cases of creating images.

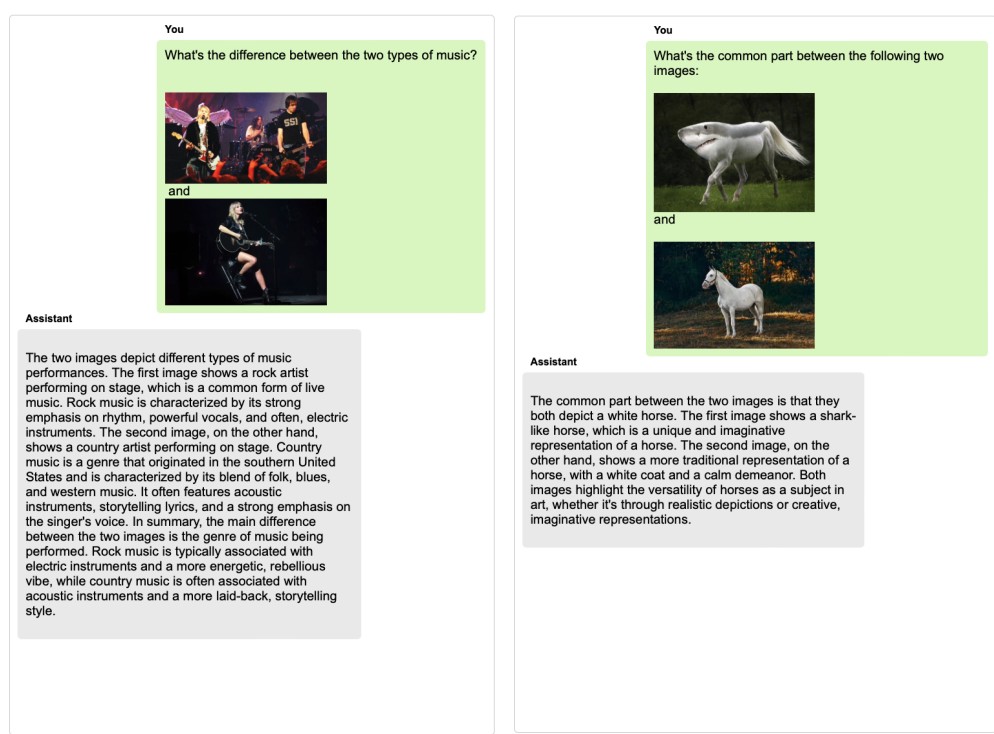

(a) Comparing music styles.    (b) Relating images.

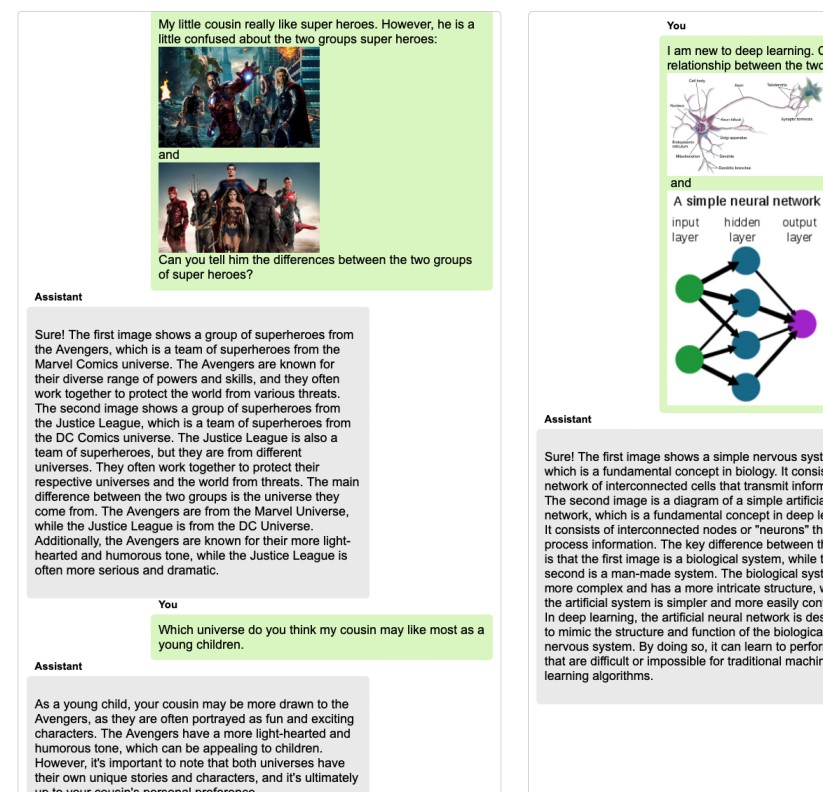

(c) Comparing movies.    (d) Comparing different concepts.

Figure 6: User cases of comparing images.

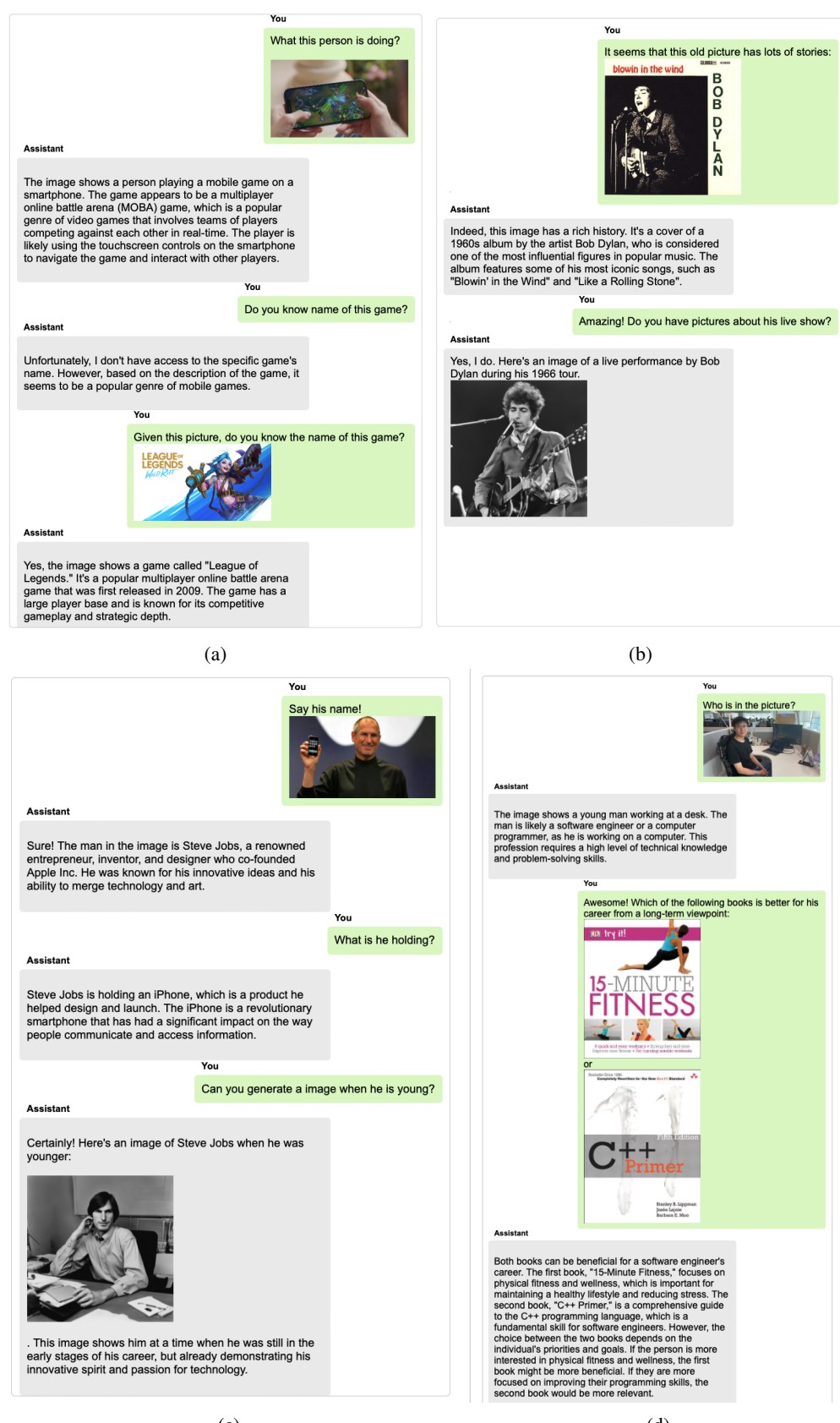

Figure 7: User cases of understanding both intrinsic & extrinsic information in the images.

## B    PROMPT OF TEXTBIND

---

**GPT-4 Prompt**

Please construct a dialogue between a human and a helpful, honest and harmless assistant. The dialogue contains interleaved text and images. Each image is represented by <imgX> DESCRIPTION </imgX>, where DESCRIPTION is a textual description of the image and X is an index of the image. Please do not assume any further visual information beyond the description.

The constructed dialogues must and can only contain the following input images:
<img0> museum - the 1st nuclear submarine </img0>
<img1> response to the production of heavy </img1>

**Characteristics about the assistant:**
1. The assistant is trained to understand text, images, and their combinations.
2. The assistant can reply the human with images and/or text.
3. The assistant has exceptional world knowledge and commonsense reasoning capabilities.
4. The assistant does not have access to the Internet or any other external tools.
5. If the assistant is asked to create an image, it can only show the image in the provided image list.
6. Please do not copy the images appearing in the dialogue. The assistant should refer to the previously mentioned image by natural language.

**Characteristics about the human:**
1. The human may send images and/or text to the assistant.
2. The human may ask questions requiring visual reasoning and/or understanding the relations between multiple images.
3. The human may ask the assistant to show images based on his/her intention.
4. The human may ask the assitant to do interesting things, rather than simply describing the content of the image.

**Properties of a bad dialogue:**
1. Simply describing or analyzing the content in the image.
2. Dialogue without a good logic.

**Properties of a good dialogue:**
1. Introducing extrinsic and attractive information of stuff shown in the image.
2. Discovering the connection between multiple images.
3. The dialogue happens in a reasonable context.

**Examples:**
========
Case 1:
Input Images: . . .
Output Dialogue: . . .
______
Case 2:
Input Images: . . .
Output Dialogue: . . .
______
Case 3:
Input Images: . . .
Output Dialogue: . . .
======

Please directly give the dialogue if you understand. The number of turns of the dialogue should be less than 6. The dialogue should be self-contained. Do NOT assume any previous dialogue between the human and the assistant. Please use the same format <imgX> DESCRIPTION </imgX> to denote images in the dialogue and do not modify the description of the image. Note that the dialogue can stop at any time without an explicit ending.

---

Figure 8: Illustration of the prompt used for TEXTBIND.

## C  EXAMPLES OF CONSTRUCTED CONVERSATIONS

---

**GPT-4 Ouput**

**Example #1**

**Human**: Hi! I'm trying to write **a cartoon story** for my kids featuring a grumpy clown and a lovable cupcake. Can you help me craft this?

**Assistant**: Of course! That sounds like a wonderful idea. We can start by establishing our characters. The grumpy clown could be someone who struggles finding joy in his own performances. <img0> a cartoon illustration of a clown looking angry </img0>

**Human**: That's a wonderful idea! How about **the cute cupcake**?

**Assistant**: The cupcake, on the other hand, is always upbeat and tries to help the clown find his happiness. Despite being a tiny cupcake, it has a big heart and a contagious joy. <img1> cartoon illustration of a cupcake with a happy expression </img1>

**Human**: Interesting! What should be their adventure?

**Assistant**: They could go on a journey to find the ultimate joke that guarantees laughter. The lovable cupcake, with his relentless positivity and cheer, helps the grumpy clown see the lighter side of life and finally find his joy.

**Example #2**

**Human**: Hi, I am researching small business operations. Can you show me an example of informal small business?

**Assistant**: Of course. Here's an example of informal small business: <img0> a mother and daughter selling gum and cigarettes in person </img0>. This concept of business is most commonly found in rural or underdeveloped areas where formal establishments might not be as prevalent.

**Human**: Interesting! On a somewhat related note, what sells well in such arrangements?

**Assistant**: It depends on the location and the demand. Mostly, such businesses sell products of daily use such as food items, candy, toiletries etc. For example an ice cream cart is a common sight in many areas, as it's a popular treat among people of all ages.

**Human**: Can you show me an image of a kid indulging in something like that?

**Assistant**: Absolutely! Here's a delightful image of <img1> a child eating ice cream </img1>. Ice cream and similar snacks are often popular products for small informal businesses.

**Example #3**

**Human**: Hi there! I'm preparing a speech on unity and cooperation in global issues. I have a few images but I'm not sure how to describe them in a way that fits my theme.

**Assistant**: I'd be happy to help with that! Could you show me the images?

**Human**: Sure, here they are. <img0> large group of people in the shape of flag </img0> and <img1> rear view of a male boxer holding globe with flag painted on his back </img1>

**Assistant**: The first image can be seen as a symbol of unity. Here, numerous individuals come together to form the shape of a flag, showing that collective action can lead to a powerful representation, like a flag - often a symbol of identity and unity. The second image illustrates the idea of global responsibility and commitment. The male boxer, suggesting strength and determination, has a globe – representing the world – in his hands, which could indicate a keen sense of global responsibility. The flag painted on his back can symbolize allegiance or commitment to his country, and by extension to global harmony if we're expanding the theme to encompass international cooperation.

**Human**: That sounds great! I also have this image. Not sure if it fits. <img2> diplomatic handshake between countries : flags overprinted the hands stock photo </img2>

**Assistant**: This image very much fits your theme. It can represent a formal agreement or cohesion between countries. The flags overprinted on the shaking hands signify the countries involved in the meet. This image can be used to illustrate diplomatic dialogue or mutual understanding which is an integral component to solving global issues through cooperation and unity.

---

Figure 9: Illustration of in-context examples of TEXTBIND.

## D  COMPARISON WITH PREVIOUS DATASETS.

We further investigate the lexical diversity of generated data, including both instructions and responses. The diversity of text in a dataset is defined as $\sum_{n=2}^{4}(\frac{\#\text{Unique n-grams}}{\#\text{Total n-grams}})$, in line with previous works (Su

| Dataset | Multi-turn | Arbitrarily Interleaved | Multimodal | Instruction Following | Image Generation |
|---|---|---|---|---|---|
| PLATYPUS | ✗ | ✗ | ✗ | ✓ | ✗ |
| FLAMINGO | ✗ | ✓ | ✓ | ✗ | ✓ |
| MULTIINSTRUCT | ✗ | ✗ | ✓ | ✓ | ✗ |
| MINIGPT-4 | ✗ | ✗ | ✓ | ✓ | ✗ |
| LLAVA | ✓ | ✗ | ✓ | ✓ | ✗ |
| TEXTBIND | ✓ | ✓ | ✓ | ✓ | ✓ |

Table 8: Comparison with previous datasets.

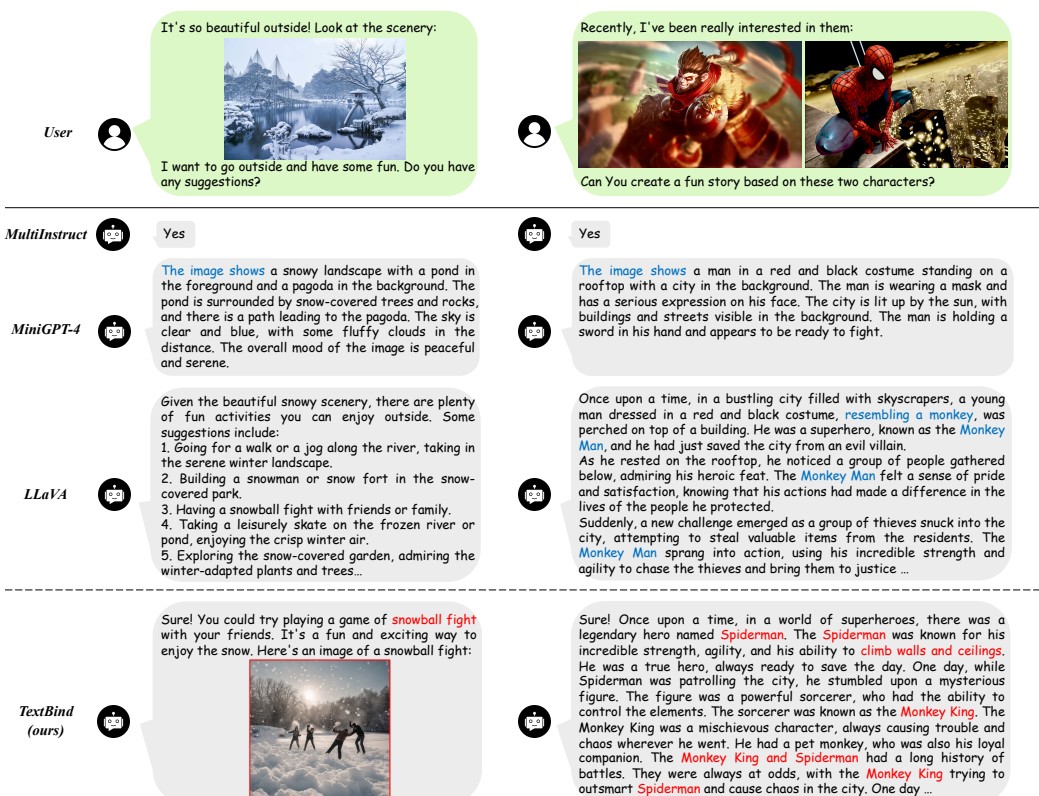

Figure 10: Comparison among models trained on different datasets.

et al., 2022). As shown in Tab. 2, our dataset achieves better diversity scores than most existing visual-language datasets, indicating that the language used in our dataset is more informative.

# E  HUMAN ANNOTATION GUIDELINE.

The comprehensive guideline for human evaluation is shown in Table 9.

# F  IMPLEMENTATION DETAILS (DATA)

We construct our TEXTBIND dataset based on the CONCEPTUAL CAPTIONS 3M (CC3M) (Sharma et al., 2018; Changpinyo et al., 2021) dataset, which only provides image-caption pairs. In our experiments, we employ the clip-vit-base-patch16 model released by OpenAI[7] (Radford et al., 2021) to filter out image-caption pairs with matching scores lower than 30. We use the $k$-means clus-

---

[7]https://huggingface.co/openai/clip-vit-base-patch16

| Annotation | Labels | Description |
|---|---|---|
| Overall Quality | Excellent | This conversation is very interesting, practical, or intricate. |
| | Satisfactory | This conversation is coherent and reasonable without any factual errors. |
| | Poor | At least one turn in the conversation is unreasonable in some aspects, e.g., unrealistic content, illegal formats, etc. |
| Required Abilities | Image Creation | To create new images in appropriate contexts. |
| | Image Comparison | To combine, relate, or compare the information in different images. |
| | Intrinsic Image Understanding | To identify and recognize the objects, colors, shapes, and patterns in images. |
| | Extrinsic Image Understanding | To interpret the underlying meaning of images, e.g., the context, emotions, symbolism, or narrative conveyed by the images. It goes beyond merely recognizing the elements in the images and often requires external knowledge and/or deep analysis. |

Table 9: Human annotation guideline.

| Training Stage | Epoch | Learning Rate | Batch Size | Max Sequence Length | Training Modules |
|---|---|---|---|---|---|
| Multimodel Alignment | 2 | 1e-4 | 256 | 256 | Q-Former, Linear |
| Multimodel Instruction Following | 3 | 1e-5 | 64 | 768 | Q-Former, Linear, LLM |

Table 11: Training Configures of our Experiments

tering algorithm implemented by FAISS (Johnson et al., 2019) toolkit to classify the cleaned CC3M dataset into 4096 clusters. The features used for $k$-means clustering are the hidden representations of images encoded by clip-vit-base-patch16 model. In addition, clusters with less than $32$ images are regarded as outliers and will not be considered. The number of images in each conversation is sampled from $\{2, 3, 4\}$. We access the GPT-4 model through the OpenAI API[8], and set top_p and temperature hyper-parameters to $1.0$.

## G  CONSTRUCTED CONVERSATIONS WITH "POOR" LABEL

In Table 10, we identify three typical errors present in the constructed dataset. Despite setting a high threshold to filter out mismatched image-caption pairs, some mismatched cases cannot be detected by the CLIP model (Radford et al., 2021). A few conversations suffering from incoherence and hallucinations may be attributed to the GPT-4 model. Overall, while a small number of conversations are affected by errors that are difficult to detect using rules, most generated conversations exhibit high quality. We present several cases labeled with "Poor". We can find that most of those "Poor" cases only have minor and non-obvious problems.

## H  IMPLEMENTATION DETAILS (MODEL)

Our experiments are based on Huggingface Transformers[9] (Wolf et al., 2020) and DeepSpeed[10] (Rasley et al., 2020). We use the filtered synthetic captions given by BLIP[11] (Li et al., 2022), including Conceptual Captions (Changpinyo et al., 2021; Sharma et al., 2018) and SBU (Ordonez et al., 2011), totally 12M image-caption pairs. We employ the same visual encoder and Q-Former as used in BLIP-2 (Li et al., 2023b) and use their weights for initialization. LLama2-Chat[12] (Touvron et al., 2023)

| Error Type | Percentage |
|---|---|
| Img-Cap Mismatch | 0.03 |
| Incoherence | 0.03 |
| Hallucination | 0.03 |

Table 10: Error types and percentage in dataset constructed by TEXTBIND.

is utilized as the backbone language model. For the image generation model, we use Stable Diffusion

---

[8]https://openai.com/blog/openai-api

[9]https://huggingface.co/docs/transformers/index

[10]https://github.com/microsoft/DeepSpeed

[11]https://github.com/salesforce/BLIP

[12]https://huggingface.co/meta-llama/Llama-2-7b-chat-hf

XL[13] (Podell et al., 2023). The training configures are shown in Table 11. We use 8 NVIDIA A100 (40G) GPUs for all experiments.

## I   HUMAN EVALUATION GUIDELINE

- Score 4: The response is excellent.
- Score 3: The response is acceptable but may not be very informative and interesting.
- Score 2: The response has minor issues, such as slight hallucinations when describing the images in the context.
- Score 1: The response is invalid and has significant drawbacks, e.g., irrelevant to the context.

## J   DATA FORMAT OF MODEL INPUT

```
{
    "conversation" {
        {
            "role": "user",
            "content": "<image> Do you...",
            "image_list": ["1235.png"]
            "caption_list": ["a runing husky ..."]
        },
        {
            "role": "assistant",
            "content": "Yes, I do! ...",
            "image_list": [],
            "caption_list": []
        },
        ...
    }
}
```

---

[13]https://github.com/Stability-AI/generative-models

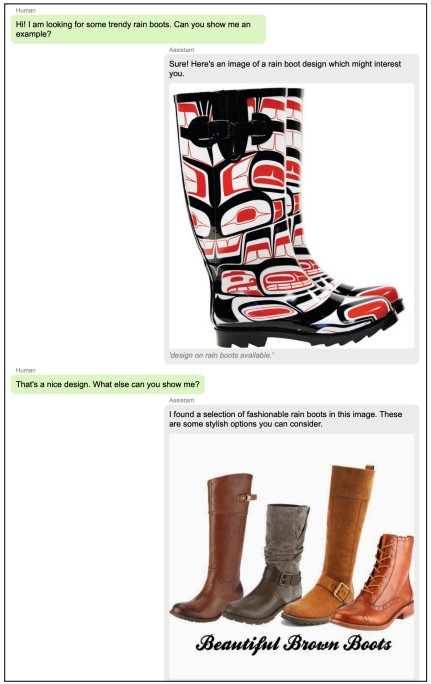

(a) Hallucination Case: The topic is about rain boot. However, not all the boots shown in the second image are rain boots.

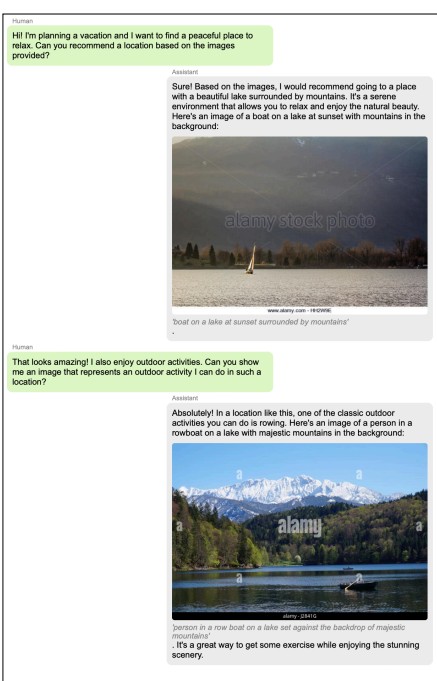

(b) Incoherence Case: The first turn discusses about relaxing while the second turn is about outdoor activities. There are no connections.

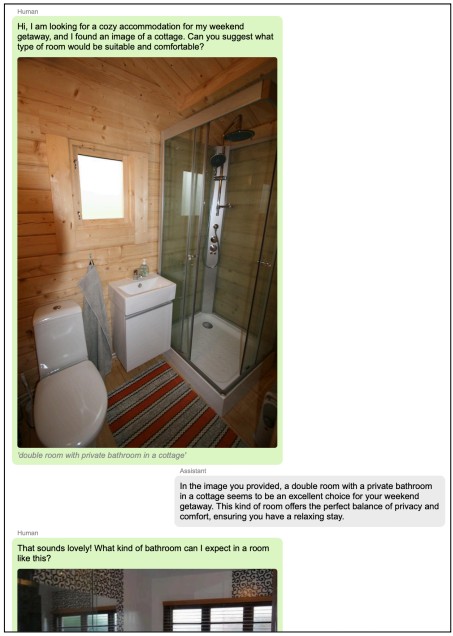

(c) Image-caption Mismatch Case: The first image only shows a bathroom, but the caption is "double room with private bathroom in a cottage".

Figure 11: Constructed conversations with "Poor" Label. The caption is shown below the image with gray color.

