# OpenReview forum: "TextBind: Multi-turn Interleaved Multimodal Instruction-following in the Wild"
_ICLR.cc/2024/Conference — Submitted to ICLR 2024_

### Official Review · Reviewer_TZuK · 2023-10-26

**Soundness:** 2 fair
**Presentation:** 3 good
**Contribution:** 3 good
**Rating:** 6
**Confidence:** 3

**Summary:**

This work proposes a pipeline to automatically construct multi-turn multimodal conversation-like instruction tuning datasets for models to be trained and evaluated on. The pipeline first ensembles a group of images with similar targeted concepts and interleaves them with an LLM to construct a multimodal conversation. A post-refinement stage is performed to ensure the quality, with a golden seed set serving as the in-context candidate for the LLM generations.
The authors then propose an MIM training to incorporate the curated data, with representations from stable diffusion models, for augmenting LMs with the ability to interleave images between textual conversations.

**Strengths:**

- The proposed multimodal instruction curation pipeline is sound, and the steps are claimed to incorporate quality assurances.
- The proposed MIM method should suit a wider range of multimodal conversational tasks.
- The experiments are compared with some strong recent multimodal models on the instruction following regime. (Although at the point of review, GPT-4V is available.)

**Weaknesses:**

- Although I agree with the rationale behind relatively poor performance from TextBind to other frameworks on the existing benchmarks, the proposed data curation framework should be aimed at such a generality, and hence leaves a drawback here where it cannot deal with some lower-level vision tasks (which are still quite important). What I would like to see is, could the models (MIM here) benefit further from tuning on these low-level vision instructions too, and at least achieve an on-par performance with existing works. That would strengthen the work significantly.
- The criteria of the conversation might be too coarse and not informative. The criteria seems holistic but in Section 4 it is said per-turn utterance is inspected, too. How do they correlate?  Also, has the annotator agreement been computed and reported? How many are there and what are their relations to the research group?
- A study of training conversations for tuning against the model performance is needed to gauge the contribution of the data.

**Questions:**

- Any analysis on what kinds of conversion utterance is likely leading to an image, but failed to generate from the models, and vice versa?

**Details Of Ethics Concerns:**

None.

---

> ### Author Response · Authors · 2023-11-20
> **Response to Reviewer TZuK (Part 1)**
>
> Thanks for your positive feedback and insightful questions. We appreciate that you think our method to create the dataset is “sound”, and the proposed model is flexible and effective enough to “suit a wider range of multimodal conversational tasks”. Below, we would like to answer your concerns point-by-point.
>
> ***Q1: Although I agree with the rationale behind relatively poor performance from TextBind to other frameworks on the existing benchmarks, the proposed data curation framework should be aimed at such a generality, and hence leaves a drawback here where it cannot deal with some lower-level vision tasks (which are still quite important). What I would like to see is, could the models (MIM here) benefit further from tuning on these low-level vision instructions too, and at least achieve an on-par performance with existing works. That would strengthen the work significantly.***
>
> We would like to highlight that the capbilities that TextBind dataset can bring to the model is almost orthogonal to conventional multimodal instruction-following datasets. According to the results in Table 6, we find that mixing TextBind with other datasets (the last row labeled with “Mix”) achieves substantial better overall performance on MMBench.
>
> ***Q2: The criteria of the conversation might be too coarse and not informative. The criteria seems holistic but in Section 4 it is said per-turn utterance is inspected, too. How do they correlate? Also, has the annotator agreement been computed and reported? How many are there and what are their relations to the research group?***
>
> Due to the page limit, the detailed guideline of the human annotation is shown in Table 8 (Appendix E) of the paper. If any turn of the conversation is unreasonable in some aspects, e.g., unrealistic content, illegal formats, etc, the whole conversation will be labeled as “Poor” quality. If there is no errors in any turn of the conversation, the conversation will be labeled as “Satisfactory”. If the whole conversation is very interesting, practical, or intricate, we will annotate it as “Excellent”.
>
> Three authors of this work are asked to annotate the data, following [1]. The main reason is that the annotation of the multimodal conversation is very challenging yet the total number of annotated cases are relatively small. Note that the above annotation is for training data construction. For evaluating performance, we use a third-party annotation platform and report  inter-annotation agreement in Table 4.
>
> [1]: Self-Instruct: Aligning Language Models with Self-Generated Instructions. ACL 2023.
>
> ***Q3: A study of training conversations for tuning against the model performance is needed to gauge the contribution of the data.***
>
> Thanks for your valuable suggestion! To address your question, we conducted an in-depth analysis to show where the derived dataset helps.  In addition to the results in Table 5, we ask human annotators evaluate the quality of the generated responses in terms of three fine-grained dimensions: instruction-following (fulfill the user’s intent), multi-modal context understanding (correctly understand the information in text and images), and the informativeness of the generated responses. For each dimension, a human annotator will assign a score in {1,2,3,4}. The four scores ranging from 1 to 4 indicate “major error”, “minor error”, “acceptable”, and “perfect”, respectively. We compare TextBind with LLaVA (the second best model in our holistic evaluation in Sec. 6.4) on 100 randomly sampled data. As shown in the following table, the model trained on TextBind can better follow the instructions of humans and leverage the multi-modal context. Notably, the informativeness of model trained on TextBind is comparable with that trained on LLaVA.
>
> |          | Instruction-following | Multi-modal Context Understanding | Informativeness |
> |----------|-----------------------|-----------------------------------|-----------------|
> | LLaVA    | 3.59                  | 3.56                              | 3.78            |
> | TextBind | 3.99                  | 3.82                              | 3.72            |
>
> **(Unfinished)**

---

> > ### Author Response · Authors · 2023-11-20
> > **Response to Reviewer TZuK (Part 2)**
> >
> > ***Q4: Any analysis on what kinds of conversion utterance is likely leading to an image, but failed to generate from the models, and vice versa?***
> >
> > Thanks for your insightful question. There are two kinds of conversation contexts that can lead to image generation. First, the conversation context has an explicit or implicit dictation for image generation (e.g., the last utterance is “could you show me an image of …”). Second, it is also appropriate to generate an image even without such an explicit/implicit dictation. For example, in Figure 6 (Appendix A), the model trained on TextBind can explain the difference between football and soccer using images spontaneously.
> >
> > For the first category, we find our model can correctly follow the explicit dictations for image generation in almost all cases. For the second category, either replying with text or images is acceptable. Sometimes the golden-truth may contain images but the model only produce textual response.

---

> > > ### Comment · Reviewer_TZuK · 2023-11-21
> > >
> > > Thanks for the additional results and analysis, I think they make the paper more solid.
> > >
> > > **For Q4:**
> > > I think the first category is obvious, what I care about more is the second one. What I'm looking for is either a quantitative (or rich qualitative) results showing that, there are some turns in a conversation a human would expect providing images would be more helpful, and that the model decides not to. I'd like to see what those cases are.
> > >
> > > Nevertheless, I was already leaning towards positive, and I remain my stand. Thanks!

---

> ### Author Response · Authors · 2023-11-22
> **Follow-up on the new comments**
>
> Dear Reviewer TZuK,
>
> Thanks for your valubale comments. We are very happy that the previous response addressed most of your concerns. We will include more case demonstrations for the cases that you are interested in. Thank you again!

---

### Official Review · Reviewer_bpnq · 2023-10-30

**Soundness:** 2 fair
**Presentation:** 3 good
**Contribution:** 2 fair
**Rating:** 5
**Confidence:** 4

**Summary:**

This paper introduces TEXTBIND, an almost annotation-free framework for empowering LLMs with multi-turn interleaved multimodal instruction-following capabilities. It devises an automatic data generation method with image clustering and GPT-4 generation to obtain a new multi-turn interleaved multi-model instruction dataset. It devise MIM, to integrate image encoder and decoder models for both image-grounded text generation and image generation. Experiment shows the effectiveness of the method and new derived dataset.

**Strengths:**

1. It devises an almost annotation-free framework for empowering LLMs with multi-turn interleaved multimodal instruction-following capabilities, and provides a multi-turn interleaved image-text dataset.
2. It proposes a method to support both image-grounded dialogue generation and image generation, and the experiment validates the effectiveness of training on the provided dataset for multimodal instruction-following.

**Weaknesses:**

1. The evaluation is mainly based on the derived TEXTBINDEVAL dataset, and also seems do damage to the results of the benchmark datasets. Besides, given the automatic text generation metrics such as BLEU, Rouge, it is hard to know on where the derived dataset and training helps.
2. For the data collection procedure, clustering the image together and chat about the visually similar images may not be the real-world demands for multimodal instruction-following.
3. Compared with other work about multimodal instruction-following, such as Otter[1], what is the advantage and difference of this method?

[1] Otter: A Multi-Modal Model with In-Context Instruction Tuning

**Questions:**

1. For the data collection procedure, how to decide the number of images to chat in a conversation and where to insert the image input or response?
2. Where does the method improve for the common benchmarks or whether the author has tested its performance on other interleaved or in-context datasets such as MIMIC-IT [2]?

[2] MIMIC-IT: Multi-Modal In-Context Instruction Tuning

---

> ### Author Response · Authors · 2023-11-20
> **Response to Reviewer bpnq**
>
> Thanks for your detailed and constructive review. We would like to address your concerns as follows.
>
> ***Q1: The evaluation is mainly based on the derived TEXTBINDEVAL dataset, and also seems do damage to the results of the benchmark datasets. Besides, given the automatic text generation metrics such as BLEU, Rouge, it is hard to know on where the derived dataset and training helps.***
>
> We would like to first clarify that we also conduct evaluation on two popular multimodal benchmarks, i.e., MME and MMBench. We also want to emphasize that the ability that TextBind dataset can bring to the model is almost orthogonal to conventional multimodal datasets, e.g., VQA and captioning. The TextBind is designed to enable the model to have open-world conversations with humans. It is not surprise that the performance of the model solely trained on TextBind cannot achieve very good performance on conventional multimodal benchmarks. However, this problem can be substantially alleviated by simply mixing the data of TextBind with other datasets. According to the results in Table 6, we find that mixing TextBind data with other datasets achieves much better overall performance on MMBench.
>
> We very much appreciate your suggestion on conducting an in-depth analysis to show where the derived dataset and training helps.  Following your suggestion, in addition to the results in Table 5, we ask human annotators evaluate the quality of the generated responses in terms of three fine-grained dimensions: instruction-following (fulfill the user’s intent), multi-modal context understanding (correctly understand the information in text and images), and the informativeness of the generated responses. For each dimension, a human annotator will assign a score in {1,2,3,4}. The four scores ranging from 1 to 4 indicate “major error”, “minor error”, “acceptable”, and “perfect”, respectively. We compare TextBind with LLaVA (the second best model in our holistic evaluation in Sec. 6.4)  on 100 randomly sampled data. As shown in the following table, the model trained on TextBind can better follow the instructions of humans and leverage the multi-modal context. Notably, the informativeness of model trained on TextBind is comparable with that trained on LLaVA.
>
>
> |          | Instruction-following | Multi-modal Context Understanding | Informativeness |
> |----------|-----------------------|-----------------------------------|-----------------|
> | LLaVA    | 3.59                  | 3.56                              | 3.78            |
> | TextBind | 3.99                  | 3.82                              | 3.72            |
>
>
> ***Q2: For the data collection procedure, clustering the image together and chat about the visually similar images may not be the real-world demands for multimodal instruction-following.***
>
> We would like to clarify that the clustered images are not visually similar, but semantically relevant. We use the hidden states got from CLIP instead of the pixel information to cluster images. As shown in Figure 2, the images belonging to the same cluster share very similar semantics. We believe that, in order to facilitate coherent, meaningful, and practical conversations, the images within each group should exhibit meaningful interconnections.
>
> ***Q3: Compared with other work about multimodal instruction-following, such as Otter[1], what is the advantage and difference of this method?***
>
> The research problem of Otter[1] and ours are quite different. Otter follows the research line of Flamingo, which focuses more on the in-context learning (or few-shot learning). The format of each in-context example of Otter  is <instruction, image, answer>, and the Otter model is trained to provide a better answer given the <query instruction, query image> and multiple in-context examples. In contrast, our model focuses on the open-word multi-turn multi-modal conversation between a human and an assistant.
>
> ***Q4: For the data collection procedure, how to decide the number of images to chat in a conversation and where to insert the image input or response?***
>
> As mentioned in Appendix F, the number of images for each conversation is randomly sampled from a uniform distribution [2, 4]. We let the GPT-4 to deice the positions of the images appearing in the conversation. In Figure 3(c), we report the distribution of the image positions.
>
> ***Q5: Where does the method improve for the common benchmarks or whether the author has tested its performance on other interleaved or in-context datasets such as MIMIC-IT [2]?***
>
> The MIMIC-IT dataset is an in-context learning (few-shot learning) dataset with <instruction, image, answer> instances, which does not fit our setting. In addition, MIMIC-IT is only a training dataset, and does not provide an evaluation set.

---

> > ### Author Response · Authors · 2023-11-22
> > **Follow-up on Our Previous Response**
> >
> > Many thanks for your valuable time and thoughtful review. As the discussion period is coming to an end, we want to check in and see if our previous response has addressed your concerns. If you have additional questions or concerns for raising the rating, please do not hesitate to reach out to us! Thank you again!

---

### Official Review · Reviewer_1Usx · 2023-10-31

**Soundness:** 2 fair
**Presentation:** 2 fair
**Contribution:** 2 fair
**Rating:** 5
**Confidence:** 5

**Summary:**

This paper explores interleaved multimodal conversations with LLMs. The authors first construct a multimodal conversation instruction tuning dataset with text-only GPT-4 and image descriptions. Then, multi-turn interleaved multimodal (dubbed MIM) instruction tuning based on GILL is used to train the multimodal LLMs to learn which segment of text information should be used for diffusion model image synthesis (which also defines when to generate images). Experiments demonstrate promising results.

**Strengths:**

- The targeted problem of enabling multimodal LLMs to generate images for a multimodal multi-turn conversation is trending, interesting, and important. Great application potentials can be induced in both the research community and the industry.
- The proposed dataset would be very useful to the community if it is open-sourced. The construction involves human-in-the-loop refinement, which is good since the data from the internet is extremely noisy.

**Weaknesses:**

- **Major concern 1.** My first major concern lies in the novelty of the proposed textual exchange method for enabling LLMs to generate images. As far as I know, learning which segment of texts to use as the text inputs of text-to-image models was first proposed by Divter [1] (not cited or discussed). Divter proposes to learn the textural inputs by using a special token and constructed template. However, leveraging LLMs and diffusion models is different, but the contribution is still limited in this case. The authors are required to clarify the similarities and differences with Divter. On the other hand, using only textual conditions to generate images has its limitations when considering very long-context-based (e.g., interleaved documents) image generation or image-conditioned image generation (e.g., image2image translation, image edition, etc).
- **Major concern 2.** My second major concern lies in the experimental evaluations. i) In Table 3, more commonly used NLP benchmarks like MMLU, HellaSwag, and WinoGrande should be conducted. In Table 5, what about the most commonly used metric FID results? In Table 6, I wonder about the zero-shot results of TextBind on VQAv2 and MM-Vet. ii) No ablation studies or in-depth discussions are presented. For example, what if no human-in-the-loop is used during dataset construction? What are the failure cases of TextBind, and why? What emerging properties could be explored by TextBind?
- **Technical contribution.** The proposed method is simple, but the technical contribution is limited. The proposed multimodal LLM architecture is mainly based on previous work GILL. The only difference is the textural information exchange method which is good but somewhat incremental. Besides, the topic-aware image sampling is quite similar to the dataset constructed by PALI-X [2]. PALI-X constructs its own interleaved dataset Episodic WebLI by grouping image-text pairs.
- **Unclearly supported claims.** The authors claim that the current multimodal instruction tuning methods lead to limited performance in open-world scenarios. However, there is a lack of analysis of TextBind's superiority in it. Besides, annotation-free may be overclaimed since human-in-the-loop definitely requires non-trivial annotation efforts.

[1] Multimodal Dialogue Response Generation. In ACL 2022.
[2] PaLI-X: On Scaling up a Multilingual Vision and Language Model.

**Questions:**

- Why is the method called TextBind? Assuming the authors are trying to analog to ImageBind [3]. However, the core spirit of multimodal binding in the same embedding space as a modality-agnostic multimodal encoder is very different from this paper. I am quite confused about this. A name that better summarizes the work's idea is better than using one similar to an existing work while not very suitable.
- Will the curated dataset be released to the public?
- The dataset is constructed by using image descriptions with GPT-4, similar to LLaVA [4] and ChatCaptioner [5] (not cited). I wonder how is the hallucination problem of data and models in such progress since CLIP filtering can not guarantee the avoidance of such an issue. For example, can authors provide some failure cases of the dataset and test the model's hallucination capability?
- Now we have GPT-4V, I am looking forward to the newly constructed dataset with GPT-4V (not required at this moment).
- Since Q-Former is used, which may compress the visual signals, I wonder about the OCR performance of TextBind. For example, zero-shot results on TextVQA?
- There are many concurrent works working in this direction. It would be good if these works were discussed in related work [6-10].
- Minor: When the abbreviation `MIM` first appears in the paper, there is no explanation of the meanings.

I am looking forward to the authors' response.

[3] ImageBind: One Embedding Space To Bind Them All. In CVPR 2023.\
[4] Visual instruction tuning. In NeurIPS 2023.\
[5] ChatGPT Asks, BLIP-2 Answers: Automatic Questioning Towards Enriched Visual Descriptions. arXiv 2023.\
[6] Generative Pretraining in Multimodality. arXiv 2023.\
[7] DreamLLM: Synergistic Multimodal Comprehension and Creation. arXiv 2023.\
[8] MiniGPT-5: Interleaved Vision-and-Language Generation via Generative Vokens. arXiv 2023.\
[9] NExT-GPT: Any-to-Any Multimodal LLM. arXiv 2023.\
[10] Kosmos-G: Generating Images in Context with Multimodal Large Language Models. arXiv 2023.

---

> ### Author Response · Authors · 2023-11-20
> **Response to 1Usx (Part 1)**
>
> Thanks for your detailed and constructive review and acknowledging our work as “interesting”, “important”, and “very useful”. In the following, we would like to answer your concerns one-by-one.
>
> ***Major concern 1:  My first major concern lies in the novelty of the proposed textual exchange method for enabling LLMs to generate images. As far as I know, learning which segment of texts to use as the text inputs of text-to-image models was first proposed by Divter [1] (not cited or discussed). Divter proposes to learn the textural inputs by using a special token and constructed template. However, leveraging LLMs and diffusion models is different, but the contribution is still limited in this case. The authors are required to clarify the similarities and differences with Divter. On the other hand, using only textual conditions to generate images has its limitations when considering very long-context-based (e.g., interleaved documents) image generation or image-conditioned image generation (e.g., image2image translation, image edition, etc).***
>
> Thanks for pointing out the missing reference. We agree that Divter also generates images from the image descriptions, and we will add the following discussion to the next version of our paper.
>
> There are several differences we would like to highlight between our work and Divter:
>
> 1. **Objective.** Divter focuses on the chitchat scenarios, where two users may share one photo during casual conversations. In contrast, our work focuses on multimodal instruction-following between a user and an assistant.
> 2. **Data.** Each conversation used by Divter only contains one image, i.e., the shared photo. However, our work considers a more general situation, where multiple images can be dispersed throughout different conversation turns.
> 3. **Visual Input:** Divter can only accept textual input, i.e., all the input images are represented by image descriptions before inputting to the model (Sec. 4.2 [1]) . In contrast, MIM uses vision encoder (BLIP and Q-Former) to encode the images (including both human-provided and model-generated images) directly. In other words, Divert can only see image descriptions, but MIM can see real images.
> 4. **Visual Output**: Divter uses visual tokens as the medium for generating images. Nevertheless, we adopt Stable Diffusion models to generate the images. Our MIM model first generates the image description and then uses the Stable Diffusion model to translate it to an image.
>
> For your concern about the long-context-based image generation. The results in Table 5 actually show that our method achieves higher performance gains when having a longer context. Regarding the image-conditioned image generation, we agree with you that the proposed method may not perform well in this setting. This is a research direction that we are working on.
>
> ***Major concern 2: My second major concern lies in the experimental evaluations. i) In Table 3, more commonly used NLP benchmarks like MMLU, HellaSwag, and WinoGrande should be conducted. In Table 5, what about the most commonly used metric FID results? In Table 6, I wonder about the zero-shot results of TextBind on VQAv2 and MM-Vet. ii) No ablation studies or in-depth discussions are presented. For example, what if no human-in-the-loop is used during dataset construction? What are the failure cases of TextBind, and why? What emerging properties could be explored by TextBind?***
>
> Thanks for your insightful questions.
>
> For Table 3, we would like to highlight that the evaluations are focused on textual response generation in the context of **interleaved images and text**. We argue that pure NLP benchmarks are not suitable for this goal. The proposed MIM model as well as other baselines are designed for multimodal scenarios. Therefore, following almost all previous works on multimodal LLMs, we did not evaluate these models on conventional text-only NLP benchmarks. In addition, we would like to clarify that we evaluate the text output (Table 3) and image output (Table 5) generation separately because automatically evaluating the quality of the holistic interleaved text-image output is difficult (the holistic evaluation is conducted by humans and reported in Table 4).
>
> For Table 5, we have already used two popular metrics, i.e., CLIP and LPIPS, for evaluating the quality of image generation. Following your suggestion, we now report additional results of FID. As shown below, the results of FID are consistent with the other two metrics in our paper.
>
> || FID ⬇  |
> |---------------------------------------|--------|
> | SDXL| 144.76 |
> | GILL| 158.64 |
> | Q-Former as Medium| 155.01 |
> | Q-Former with Prompt Tokens as Medium | 152.23 |
> | MIM| 139.46 |
>
> **(Unfinished)**

---

> ### Author Response · Authors · 2023-11-20
> **Response to 1Usx (Part 2)**
>
> As for Table 6, its purpose is to evaluate the performance of our models on conventional vision-language tasks. We employ two popular benchmarks, MMBench and MME, because they provide a comprehensive and fine-grained evaluation. We didn't choose VQAv2 for multiple reasons. First, MMBench and MME already contain many QA tasks similar to VQAv2. Second, according to MMBench, VQAv2 uses the exact match to score model predictions, resulting in severe false-negative problem due to the open-ended nature of multimodal LLMs. MM-Vet is yet another benchmark very similar to MMBench and MME. As per your request, we conducted additional experiments on MM-Vet. The results are shown below. We can see that the results are consistent with those on MMBench and MME.
>
> || MM-Vet |
> |---------------|--------|
> | MultiInstruct | 17.2   |
> | MiniGPT-4| 9.8    |
> | LLaVA| 23.4   |
> | Shikra| 19.9   |
> | TextBind| 19.4   |
> | Mix| 23.9   |
>
> Regarding the effect of the human-in-the-loop process, we have a statistical value in the early version of the submission. This process can reduce the percentage of the “Poor” data from 30% to 9%. We identify three major causes for conversations with “Poor” label: hallucination, in-coherence, and image-caption mismatch. Discussions with statistics (Table 9) and examples (Figure 11) about those failure cases are in Appendix G.
>
> There are many emerging properties can be explored by TextBind. In Figure 5, one example is that the model trained on TextBind is able to create a new super-hero story with accurate illustrations. There are a range of capabilities of the model trained on TextBind discussed in Appendix A.
>
> ***Technical contribution. The proposed method is simple, but the technical contribution is limited. The proposed multimodal LLM architecture is mainly based on previous work GILL. The only difference is the textural information exchange method which is good but somewhat incremental. Besides, the topic-aware image sampling is quite similar to the dataset constructed by PALI-X [2]. PALI-X constructs its own interleaved dataset Episodic WebLI by grouping image-text pairs.***
>
>
> We would like to clarify the major contribution of this paper lies on the data perspective: we propose a method that can automatically construct open-world multimodal instruction-following conversations. For the model perspective, our work is indeed inspired by GILL. The only difference is that our best model uses text as medium rather than text embeddings. Although the proposed method is simple, it is well-motivated. More concretely, the proposed method aims to alleviate the one-to-many problem when generating images given a multi-turn context. To the best of our knowledge, we are the first to characterize this problem and the first to present a systematic empirical comparisons among three different variants (Section 5.1) for addressing this problem.
>
> Both the purpose and implementation of our topic-aware image sampling and the clustering in PALI-X are different. The purpose of the topic-aware image sampling in our method is to select semantically related images that can be used for constructing a meaningful and coherent multimodal conversation. However, the purpose of clustering in PALI-X is for in-context (or few-shot) learning. As for implementation, we use the CLIP model to extract the semantics of images. Nevertheless, according to Section 3.2 of PALI-X, it clustered images by URL field. In other words, they only use simple features, i.e., URL, to group some loosely related <image, text> pairs.  We will cite this PALI-X paper in the next version of this work.
>
> **(Unfinished)**

---

> ### Author Response · Authors · 2023-11-20
> **Response to 1Usx (Part 3)**
>
> ***Unclearly supported claims: The authors claim that the current multimodal instruction tuning methods lead to limited performance in open-world scenarios. However, there is a lack of analysis of TextBind's superiority in it. Besides, annotation-free may be overclaimed since human-in-the-loop definitely requires non-trivial annotation efforts.***
>
> We would like to clarify that we have conducted extensive quantitative experiments and case studies to show the TextBind's superiority in open-world scenarios. First, the TextBindEval test set constructed in Sec. 6.1 is an open-world dataset checked by humans. We did extensive experiments on the TextBindEval to show that model trained on TextBind can achieve much better performance in the open-world multimodal conversations. Second, in Appendix A, we demonstrate extensive user cases in open-world that show the capabilities of TextBind. For example, we observe that our model trained on TextBind can explain concepts and ideas for users with vivid images (Figure 6a), creating images with correct emotions (Figure 6b), and editing images based on the whole context (Figure 6c and 6d). More details about the case studies are shown in Appendix A.
>
> It seems that there is a misunderstanding about the “annotation-free”. Actually, we use “**almost** annotation-free” anywhere when we use “annotation-free”. We use “almost” because that the human-in-the-loop process indeed involves human efforts. However, the number of the human-annotated data is very small, i.e., 300 in our work compared to the total number of data we obtain (15000 and even more if we wish). We will make it clearer in the next version of this paper.
>
> ***Q1: Why is the method called TextBind? Assuming the authors are trying to analog to ImageBind [3]. However, the core spirit of multimodal binding in the same embedding space as a modality-agnostic multimodal encoder is very different from this paper. I am quite confused about this. A name that better summarizes the work's idea is better than using one similar to an existing work while not very suitable.***
>
> The name “TextBind” conveys the core idea of both the proposed data and model. For the data, we replace images with their text descriptions for multimodal conversation construction (the conversation construction is purely in the text space).  For the model, we use text as medium for image generation. In both cases, the text space is used as the transfer medium.
>
>
> ***Q2: Will the curated dataset be released to the public?***
>
> Absolutely, our dataset will be released to the public.
>
> ***Q3: The dataset is constructed by using image descriptions with GPT-4, similar to LLaVA [4] and ChatCaptioner [5] (not cited). I wonder how is the hallucination problem of data and models in such progress since CLIP filtering can not guarantee the avoidance of such an issue. For example, can authors provide some failure cases of the dataset and test the model's hallucination capability?***
>
> Thanks for the good question. Because of the page limit, the analysis of failure cases is shown in Appendix G. Note that we already reported the percentage of hallucination (Table 9) in the constructed dataset, and some detailed cases (Figure 11). We will make it clearer in the next version of this paper.
>
> ***Q4: Now we have GPT-4V, I am looking forward to the newly constructed dataset with GPT-4V (not required at this moment).***
>
> Thanks for the kind suggestion. We are also considering leveraging the visual ability of GPT-4V to create a high-quality multi-modal dataset.
>
> ***Q5: Since Q-Former is used, which may compress the visual signals, I wonder about the OCR performance of TextBind. For example, zero-shot results on TextVQA?***
>
> OCR is one of the tasks evaluated by MMBench. More concretely, in the Table 6 of our paper, the scores under the column of Fine-grained Perception (Single-Instance) (FP-S) are the evaluation results of OCR and similar tasks. The OCR performance of model solely trained on TextBind is not very good, largely because that OCR data is very rare in our TextBind dataset. However, the FP-S score can be significantly improved by simply mixing TextBind with other datasets.
>
>
> ***Q6: There are many concurrent works working in this direction. It would be good if these works were discussed in related work [6-10].***
>
> Yes, many contemporary works are released recently. We promise that we will discuss those works in the camera-ready version of this paper.

---

> > ### Author Response · Authors · 2023-11-22
> > **Follow-up on Our Previous Response**
> >
> > Many thanks for your valuable time and thoughtful review. As the discussion period is coming to an end, we want to check in and see if our previous response has addressed your concerns. If you have additional questions or concerns for raising the rating, please do not hesitate to reach out to us! Thank you again!

---

> > > ### Comment · Reviewer_1Usx · 2023-11-22
> > > **No revision?**
> > >
> > > Please first revise the paper and incorporate suggestions from all reviewers.

---

> > > > ### Author Response · Authors · 2023-11-22
> > > > **revised paper uploaded**
> > > >
> > > > Thanks for your gentle reminder. We apologize for not updating the paper at first. We have tried our best to incorporate all suggestions we received so far.  We highlighted revisions in red and marked important information in blue. Please kindly have a look and let us know if you have any further revision suggestions! We will continue to improve this paper before the camera-ready deadline!

---

### Official Review · Reviewer_E1gM · 2023-11-01

**Soundness:** 3 good
**Presentation:** 3 good
**Contribution:** 3 good
**Rating:** 8
**Confidence:** 4

**Summary:**

The paper proposes TextBind, an almost automatic and scalable pipeline for collecting multi-turn interleaved multimodal instruction-following data, and MIM, an augmented Large Language Model (LLM) with a visual encoder/decoder for supporting multimodal inputs and outputs. The MIM trained on the dataset constructed by TextBind shows a variety of visual reasoning or understanding capabilities in real-world tasks. The results of text response generation and image generation show the superiority of both the constructed dataset and the trained MIM. Overall, the quality of paper is quite good and the results are very promising.

**Strengths:**

+ The MIM seems to be the first, or at least one of the first adaptation approaches to support multimodal interleaved inputs and outputs.
+ The trained model shows superior multimodal instruction-following capabilities in real-world tasks.
+ The dataset constructed by TextBind as well as the small subset for evaluation, TextBindEval, would be very useful for future multimodal LLM research.
+ The novel collection process and the characteristics of the collected dataset are described in great detail.

**Weaknesses:**

- The paper has very good quality overall. However, it would be better to describe the architecture of MIM and the training process in more details. E.g., how is each training example constructed, that is, what is the input and the target for training the MIM model?

**Questions:**

- The collected datasets, training data and TextBindEval, would be very useful for future research. Please release them if possible.

---

> ### Author Response · Authors · 2023-11-20
> **Response to Reviewer E1gM**
>
> Thanks for your recognition of the novelty and usefulness of our work. We are happy to answer your questions as follows.
>
> ***Q1: The paper has very good quality overall. However, it would be better to describe the architecture of MIM and the training process in more details. E.g., how is each training example constructed, that is, what is the input and the target for training the MIM model?***
>
> Thanks for your valuable suggestion. We omitted some details of the implementation due to the page limit. The training examples of the MIM model follow the standard of Llama-Chat:
> ```json
> {
> 	"conversation" {
> 		{
> 			"role": "user",
> 			"content": "<image> Do you...",
> 			"image_list": ["1235.png"]
> 			"caption_list": ["a runing husky ..."]
> 		},
> 		{
> 			"role": "assistant",
> 			"content": "Yes, I do! ...",
> 			"image_list": [],
> 			"caption_list": []
> 		},
> 		...
> 	}
> }
> ```
>
> The content in different conversation turns is concatenated. The model is trained to minimize the cross-entropy loss on the assistant’s turns, conditioned on the entire preceding conversation history. Please feel free to let us know if this answer doesn't address your questions. We will make it clearer in the next version.
>
> ***Q2:  The collected datasets, training data and TextBindEval, would be very useful for future research. Please release them if possible.***
>
> We will release all the code, data, and models upon acceptance.

---

### Meta-Review · Area_Chair_T8Fi · 2023-12-15

**Metareview:**

This paper introduces a new framework, TextBind, to enhance multimodal large language models (MLLMs) with multi-turn interleaved multimodal instruction-following capabilities. This method utilizes image-caption pairs to generate multi-turn multimodal instruction-response conversations. The proposed MIM architecture integrates an image encoder and decoder, achieving better performance in multimodal conversations than baselines.

The work is commended for its novelty in adapting LLMs to multimodal interleaved inputs and outputs, one of the first works doing so. The dataset and methodology developed show superior capabilities in multimodal instruction-following.
But reviewers also had a few critical concerns.  For instance, it lacks detailed information on the MIM architecture and the training process (even after the discussion and revision). Technically, the differences between MIM and GILL are unclear.
The evaluation's focus on the TextBind dataset, and the limited performance on benchmark datasets, raise questions about the method's generalizability.

In summary, despite the timely and potentially impactful nature of the work, its current form exhibits substantial limitations.
The authors are strongly encouraged to address these critical concerns in future revisions to unlock the full potential of their proposed method.

**Justification For Why Not Higher Score:**

Same as above.

**Justification For Why Not Lower Score:**

N/A

---

### Decision · Program_Chairs · 2024-01-16

Reject